# ALICE BENCHMARKS: CONNECTING REAL WORLD RE-IDENTIFICATION WITH THE SYNTHETIC

**Xiaoxiao Sun[1], Yue Yao[1], Shengjin Wang[2], Hongdong Li[1], Liang Zheng[1]**
[1] The Australian National University    [2] Tsinghua University
{first-name.last-name}@anu.edu.au[1]  wgsgj@tsinghua.edu.cn[2]

## ABSTRACT

For object re-identification (re-ID), learning from synthetic data has become a promising strategy to cheaply acquire large-scale annotated datasets and effective models, with few privacy concerns. Many interesting research problems arise from this strategy, *e.g.,* how to reduce the domain gap between synthetic source and real-world target. To facilitate developing more new approaches in learning from synthetic data, we introduce the Alice benchmarks, large-scale datasets providing benchmarks as well as evaluation protocols to the research community. Within the Alice benchmarks, two re-ID tasks are offered: person and vehicle re-ID. We collected and annotated two challenging real-world target datasets: AlicePerson and AliceVehicle, captured under various illuminations, image resolutions, *etc*. As an important feature of our real target, the clusterability of its training set is not manually guaranteed to make it closer to the real domain adaptation test scenario. Correspondingly, we reuse existing PersonX and VehicleX as synthetic source domains. The primary goal is to train models from synthetic data that can work effectively in the real world. In this paper, we detail the settings of Alice benchmarks, provide an analysis of existing commonly-used domain adaptation methods, and discuss some interesting future directions. An online server[1] has been set up for the community to evaluate methods conveniently and fairly.

## 1 INTRODUCTION

Synthetic visual data provide an inexpensive and efficient way to obtain a large quantity of annotated data for facilitating machine learning tasks. It allows for both foreground objects and background context to be edited conveniently in order to generate diverse image styles and contents. Thanks to these benefits, synthetic data have been increasingly widely used in computer vision research and applications to improve the data acquisition process (Bak et al., 2018; Fabbri et al., 2018; Gaidon et al., 2016). Successful use cases of synthetic data include scene semantic segmentation (Hu et al., 2019; Ros et al., 2016; Lin et al., 2020; Xue et al., 2021), object detection (Gaidon et al., 2016), tracking (Fabbri et al., 2018), re-ID (Sun & Zheng, 2019; Yao et al., 2020; Wang et al., 2020; 2022; Li et al., 2021; Zhang et al., 2021; Wang et al., 2022), pose estimation (Chen et al., 2016), self-driving vehicles (Dosovitskiy et al., 2017) and other computer vision tasks (Raistrick et al., 2023b).

In this paper, we introduce a series of datasets, "Alice benchmarks" (hereafter "Alice")[2], consisting of a family of synthetic and real-world databases which facilitate research in synthetic data with applications to real-world object re-ID, including person and vehicle re-ID tasks. There are two advantages of using Alice benchmarks. Firstly, humans and vehicles are the study objects of the two tasks and will cause privacy concerns without proper de-identification (like blurring face or license plate). Learning from synthetic data offers an effective solution to this problem by decreasing the need for real-world data. Secondly, it is expensive to rely on human labor to accurately annotate a person/vehicle of interest across cameras. In comparison, using synthetic data allows us to cheaply obtain large amounts of accurately annotated images.

---

[1]details can be found at `https://sites.google.com/view/alice-benchmarks`
[2]The name "Alice" is inspired by the notion that the model learns from a synthetic environment, similar to how Alice explores the virtual world in "Alice in Wonderland".

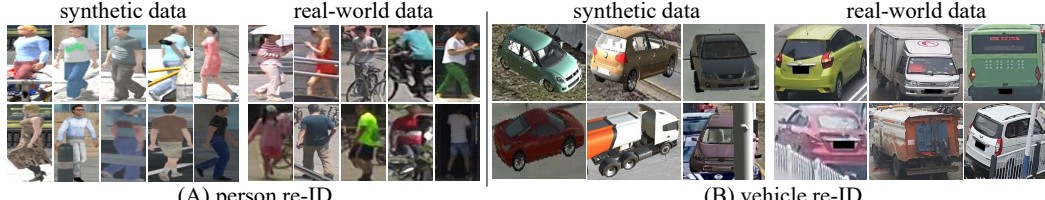

|  synthetic data | real-world data | synthetic data | real-world data |
| --- | --- | --- | --- |

(A) person re-ID    (B) vehicle re-ID

Figure 1: The current version of Alice benchmarks supports two research tasks: person and vehicle re-ID. The source domains are synthetic images from the PersonX and VehicleX datasets. The data of the target domains are real-world images we collected from varying conditions.

Specifically, Alice focuses on domain adaptation (DA) under the setting of "synthetic to real" (Syn2Real). We aim for re-ID models trained on the synthetic data to generalize well to real data. For **the source domain**, we reuse two existing synthetic datasets, PersonX (Sun & Zheng, 2019) and VehicleX (Yao et al., 2020). Specifically, the two synthetic datasets are fully editable. We combine the two engines and provide an editing interface, whereas researchers can generate their own source dataset with an arbitrary number of images under manually controllable environments. For **the target domain**, we introduce two new real-world datasets, AlicePerson and AliceVehicle. They are collected under diverse environmental conditions including resolution, illumination, scene type *etc*. Fig. 1 shows sample images. All images of those two datasets and annotations of the training sets have been released to the community[3]. Labels of the test datasets are not publicly available, and only limited online tests will be allowed in a certain period. We hope these hidden testing sets can offer a unified, fair, and challenging benchmark for Syn2Real DA research in object re-ID.

Using Alice, existing DA methods can be tested, including those using style-level alignment (Deng et al., 2018a), feature-level alignment (Long et al., 2015) and pseudo-label based methods (Zhong et al., 2020; Zheng et al., 2021; He et al., 2022). In this regard, our benchmarks offer a new platform for re-ID research. Moreover, an important advantage of Alice is that the obtained synthetic data are fully *editable*. We provide easy-to-use editing tools and user interfaces for exploring different aspects of the data, such as lighting conditions and viewpoints Some recent methods (Yao et al., 2020; 2022; Kar et al., 2019) automatically adjust the content of synthetic data, aiming to reduce the gap between the synthetic and real data. In this paper, we refer to this type of method as content-level DA.

Another key feature of the Alice benchmarks is the unassured clusterability of the target training sets. Existing domain adaptation benchmarks for re-ID typically possess a target training set with strong clusterability, *i.e.*, an individual ID often appears in multiple cameras from various viewpoints within the training set. For instance, when adapting the domain for person re-ID, the common practice is to employ the training sets of Market-1501 (Zheng et al., 2015) or MSMT17 (Wei et al., 2018). As these training sets were originally designed for effective supervised training, they typically guarantee a single ID across multiple cameras (*i.e.*, views), thus enabling robust clusterability. This characteristic facilitates pseudo-label based DA methods in achieving high scores with relative ease. However, such strong clusterability should be considered a dataset bias, as it does not often occur in real-world settings. In this study, we propose the use of AlicePerson and AliceVehicle datasets, which lack such intense clusterability. As will be demonstrated in our experiment, the weak clusterability leads to a relatively smaller accuracy improvement when pseudo-label based DA methods are employed.

To provide a baseline analysis, we evaluate some common DA methods at the content level (Yao et al., 2020; 2022), pixel level (Deng et al., 2018a), feature level (Vu et al., 2019; Tsai et al., 2018), and based on pseudo labels (Fan et al., 2018; Zhong et al., 2019; 2020). We report their accuracy not only on the Alice target datasets, but also on some existing real-world datasets, such as Market-1501 (Zheng et al., 2015) and VeRi-776 (Liu et al., 2016). With extensive experiments, we identify a number of interesting findings, such as that current state-of-the-art pseudo-label based methods are less effective on our benchmarks when the clusterability of the target domain is not guaranteed.

In addition to datasets and benchmarking, Alice provides an online evaluation platform for the community. An executable version of the system will be set up to accept submissions. Currently, two aforementioned re-ID tasks are available. In the future, we will continue building up new

---

[3]privacy information: we manually blur the human faces and license plates when applicable. Alice is distributed under license CC BY-NC 4.0, which restricts its use for non-commercial purposes.

training/testing settings for the two existing tasks and add new learning tasks. As far as we know, our server will be the first online evaluation server in the re-ID community. In summary, Alice will benefit the community in the following ways: 1) exploring the feasibility of reducing system reliance on real-world data, alleviating privacy concerns; 2) facilitating the study and comparison of DA algorithms by providing challenging datasets and a unifying evaluation platform; 3) sustaining long-term "service" for more tasks. In addition, we will discuss some potential research problems and directions made available by Alice. For example, how does "synthetic to real" compare with "real to real" regarding the domain gap problem?

This paper presents the following contributions to the computer vision community.

- We provide the Alice benchmarks, an online service for evaluating domain adaptation algorithms of object re-ID that perform learning from synthetic data and testing on real-world data. Two challenging real-world datasets have been newly collected as target sets for person and vehicle re-identification.
- We provide preliminary evaluations of some existing common domain adaptation methods on our two tasks. Results and analysis provide insights into the characteristics of synthetic and real-world data.
- We discuss some new research problems that are made possible by Alice. We also raise questions on how to better understand the Syn2Real domain adaptation problem.

## 2 RELATED WORK

**Synthetic Data**. We have seen many successful cases using synthetic data in real-world problems, such as image classification (Peng et al., 2017; Yao et al., 2022), scene semantic segmentation (Sankaranarayanan et al., 2018; Xue et al., 2021), object tracking (Gaidon et al., 2016; Yao et al., 2023b; Liu et al., 2023), traffic vision research (Geiger et al., 2012; Li et al., 2018), object re-ID (Sun & Zheng, 2019; Yao et al., 2020; Zhang et al., 2021; Wang et al., 2022) and other computer vision tasks (Raistrick et al., 2023b). For example, the GTA5 (Richter et al., 2016) dataset, extracted from the game with the same name, is well-known for domain adaptive semantic segmentation. Together with datasets like Virtual KITTI (Gaidon et al., 2016) and synthetic2real (Peng et al., 2018), they facilitate the research in style and feature-level DA. Wang et al. (2022) build the ClonedPerson dataset by cloning the whole outfits from real-world person images to virtual 3D characters. We continue this area of research by introducing Alice, aiming to further explore the potential of synthetic data.

**Domain Adaptation**. Domain adaptation is a long-standing problem, and many attempts have been made to understand the domain gap (Torralba & Efros, 2011; Perronnin et al., 2010). Recently, efforts have been made to reduce the impact of domain gap (Saenko et al., 2010; Deng et al., 2018a; Lou et al., 2019; Yao et al., 2023a). Common strategies contain feature-level (Long et al., 2015), pixel-level (Zhu et al., 2017; Deng et al., 2018b) and pseudo-label based (Fan et al., 2018; Zhong et al., 2019; Song et al., 2020b; Zheng et al., 2021; He et al., 2022) domain adaptation. Learning from synthetic data often appears together with developing domain adaptation methods. Many studies (Saito et al., 2018; Chen et al., 2019; Wang et al., 2019a; Bak et al., 2018; Yao et al., 2020; Song et al., 2020b) have been conducted investigating the Syn2Real domain gap. For example, most existing work uses domain adaptation methods (Wang et al., 2019b; Peng et al., 2019; Wang et al., 2022) to change the image style of the synthetic data to resemble that of the real-world data. On the other hand, some work (Kar et al., 2019; Yao et al., 2020; Ruiz et al., 2018) proposes a new DA strategy for adapting the content of images. In this paper, several domain adaptation strategies will be evaluated on Alice to provide a baseline analysis for object re-ID tasks.

## 3 SYNTHETIC DATA IN ALICE

The synthetic data used in Alice are generated by the existing publicly available PersonX (Sun & Zheng, 2019) and VehicleX (Yao et al., 2020) engines, used for person and vehicle re-ID, respectively.

**PersonX** has 1,266 person models that are diverse in gender (547 females and 719 males), age, and skin color. Meanwhile, the PersonX engine provides very flexible options to users, including the number of cameras, the camera parameters, scene conditions and other visual factors, such as illumination, which can be modified based on the demands of different researchers.

Table 1: Statistics of synthetic source data used in Alice benchmarks. The "arbitrary" means the number of images, cameras and scenes can be set based on the demand of users.

| datasets | # IDs/classes | # image | # scene | # camera | # camera network |
|---|---|---|---|---|---|
| PersonX & VehicleX | 1,266 & 1,362 | arbitrary | arbitrary | arbitrary | customizable |

**VehicleX** is designed for vehicle-related research. There are 1,362 vehicles of various 3D models. The environment setting is similar to PersonX. Attributes including vehicle orientation, camera parameters, and lighting settings, *etc.*, are controllable.

PersonX and VehicleX are fully editable, and some statistics of them are shown in Table 1. Here, "arbitrary" means a user can create an arbitrary number of images under user-specified environments. In other words, the settings, such as scene style, number of cameras and data format (image or video) of the dataset are customizable in those two engines. Note that while we use them to generate source data for domain adaptation in this paper, the synthetic data can also be used for other research purposes (more discussions are provided in Section E of the Appendix).

## 4    REAL-WORLD DATA IN ALICE

Alice benchmarks currently have two re-ID tasks and thus two real-world datasets, named AlicePerson and AliceVehicle, respectively, as target sets. They are collected, annotated and checked manually, and are summarized in Table 2. Details of them are described below.

**AlicePerson Dataset:** The AlicePerson dataset can be considered the successor to the Market-1501 dataset (Zheng et al., 2015). However, the settings of the AlicePerson dataset are more challenging for recognition compared to those of the Market-1501 dataset in a number of aspects: 1) there are more challenging samples (low-resolution, occlusion and illumination changes) in the new dataset, 2) the ratio of distractors is increased, such as the increases in ratios of "distractor" and "junk" in gallery, 3) the training data of AlicePerson does not have identity annotation. Note that this means each ID is not assured to appear in multiple cameras, enabling unassured clusterability.

Specifically, five cameras (three 1920×1080 HD cameras, one 1440×1080 HD camera and one 720×576 SD camera) were used during dataset collection. There are 42,760 bounding boxes (bboxes) in AlicePerson, in which 13,198 bboxes are unlabeled training data (camera IDs are available) and 29,562 bboxes are annotated validation and testing data. The validation and testing data contains a large number of "distractors". Referring to the annotation method of the Market-1501 dataset, we use Faster R-CNN (Ren et al., 2015) to detect the identities and select "good" images based on their IoU (Intersection over Union is larger than 50%) with the "perfect" hand-drawn bboxes. Moreover, "bad" detected bboxes (ratio of IoU < 20%) are labeled as "distractor". We also add some images out of the annotated identities in the gallery set as "distractor" to simulate a more realistic test situation.

**AliceVehicle Dataset:** The AliceVehicle dataset is built for vehicle re-ID and it includes data from sixteen cameras on an urban road. The main road is a straight road with some left and right branches in the middle, and the layout of the sixteen cameras is shown in Fig. 2. As can be seen, the cameras are positioned in various directions, with the fields of view (FoVs) of some cameras overlapping, such as cam8 and cam9. We kept the original distribution of the surveillance cameras on this road, such as their locations and rotations, to reduce undesired biases. All camera resolutions are 1920×1080. The training data are randomly selected from the target domain and do not have identity labels.

During annotating validation and test data, we labeled vehicles driving from cam1 to cam16 or in the opposite direction. Along the travel path of the vehicle, we gradually annotate it until it disappears from our sixteen cameras. To keep the diversity of the directions. we also labeled some vehicles from the central cameras (*e.g.*, cam8) and annotated their movement towards both sides of the road.

## 5    EVALUATION PROTOCOLS

**Source Domain.** We provide two types of synthetic data in the source domain, *i.e.,* 3D models (Table 1) in Unity and images (Table 2) captured in the virtual environment. The first type of source data is the 3D model. The provided models are fully editable and can generate an arbitrary number of images by an arbitrary number of cameras, for the two tasks. The second type of source data (the

| source: synthetic | | | |
|---|---|---|---|
| items | | PersonX | VehicleX |
| # class | | 410 | 1,362 |
| image | # camera | - | - |
| | # image | 14,760 | 45,538 |
| target: real-world | | | |
| items | | AlicePerson | AliceVehicle |
| # camera | | 5 | 16 |
| training | # image | 13,198 (31%) | 17,286 (63%) |
| | anno.? | N | N |
| | # class | N.A | N.A |
| validation | # image | 3,978 (9%) | 1,752 (6%) |
| | anno.? | Y | Y |
| | # class | 100 | 50 |
| testing | # image | 25,584 (60%) | 8,552 (31%) |
| | anno.? | Y (hidden) | Y (hidden) |
| | # class | hidden | hidden |

Table 2: Statistics summary of source and target images in Alice benchmarks.

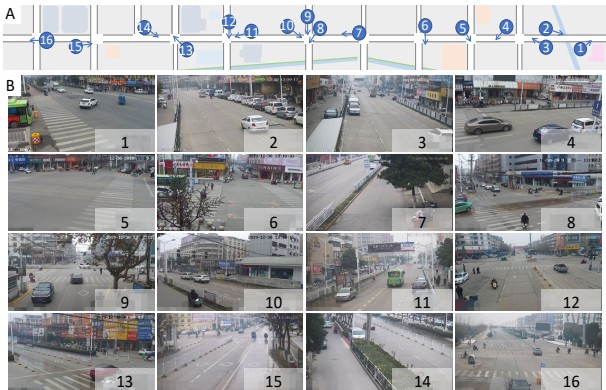

Figure 2: Camera topology in AliceVehicle. (A): Geological positions of cameras. (B): Camera field of views.

images) is given in the same way as existing synthetic datasets. These images are generated by setting environment attributes to certain values, which are randomly sampled from uniform distributions. We generated 14,760 and 45,538 images from PersonX and VehicleX engines, respectively.

**Target Domain.** For each task, we divide the real-world target data into three splits, *i.e.,* training, validation and testing (details are shown in Table 2).

The **training sets** contain 13,198 and 17,286 for the two tasks, respectively. Since we focus on unsupervised domain adaptation (UDA), the target training set should not have labels. To this end, we randomly select images from the target domain without annotating them. By *random*, we mean the number of identities for object re-ID is unknown (may contain outlier identities). Meanwhile, the number of samples per identity for object re-ID also varies—some have very few samples, and others would have a much larger number of samples. For the re-ID tasks, having an unknown number of target identities is more realistic and poses new challenges for existing algorithms. As to be demonstrated in Section 6.3, many state-of-the-art UDA methods have lower performance on AlicePerson and AliceVehicle test sets than on existing benchmarks, such as Market-1501 and VeRi-776 (Zheng et al., 2015; 2017).

Each task includes a fully annotated **validation set** for the model and its hyper-parameter selection, containing 7,978 (from 100 identities) and 1,753 (from 50 identities) images, respectively.

The **testing sets** in AlicePerson and AliceVehicle have 25,584 and 8,552 images, respectively. Their annotations are hidden and only accessible to the organizers. Meanwhile, we keep the number of IDs unknown for object re-ID tasks to make the settings to be more consistent with the real-world scenario Model performance on the test set will be calculated through our server.

**Evaluation Metric.** For object re-ID, cumulative matching characteristics (CMC) and mean average precision (mAP) are used for evaluation, which are standard practices in the community (Zheng et al., 2015; Liu et al., 2016). In CMC, rank-$k$ accuracy represents the probability that a queried identity appears in the top-$k$ ranked candidates list, and ranks 1, 5 and 10 are commonly used.

# 6 BASELINE EVALUATION

## 6.1 METHOD DESCRIPTION

**Content-level domain adaptation.** A distinct characteristic of synthetic data is that we can edit its content using the interface tools. This strategy has the potential of reducing the content gap between real and synthetic data. In this paper, we evaluate attribute descent (Yao et al., 2020), a robust and stable method in this area. Specifically, for person and vehicle re-ID, there are five attributes to be optimized, including orientation ($0^o - 359^o$), light direction (from west $0^o$ to east $180^o$), light intensity, camera height and camera distance. During learning, the attributes are modeled by single Gaussian distributions or a Gaussian Mixture Model (GMM), and the difference in data distribution

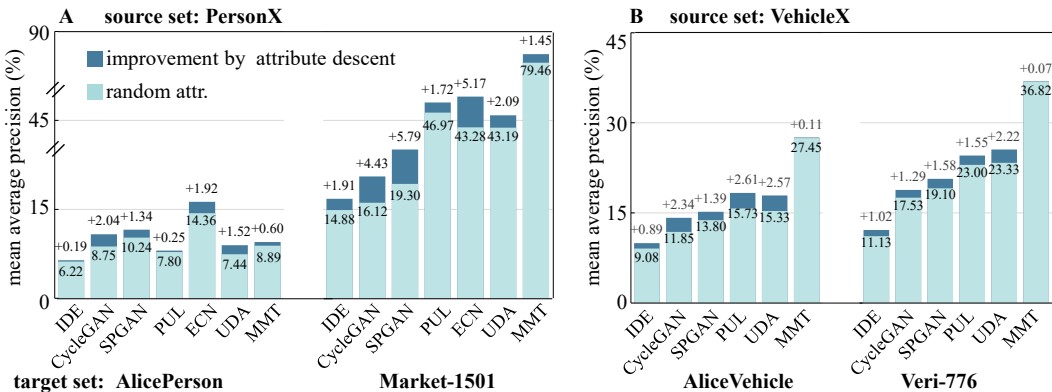

Figure 3: Performance of domain adaptation methods from synthetic to real-world datasets. **A**: person re-ID using PersonX as the source. **B**: vehicle re-ID using VehicleX as the source. Under each target dataset, we use light green to indicate the accuracy of various methods and dark green to show the improvement brought by attribute descent combined with these methods.

is measured by the Fréchet Inception Distance (FID). The objective of this optimization problem is to minimize FID between real data and synthetic data.

**Pixel-level domain adaptation.** Two methods in this direction are evaluated, *i.e.,* CycleGAN (Zhu et al., 2017) and its improvement in the re-ID task, SPGAN (Deng et al., 2018a). Specifically, CycleGAN learns two mappings, one from source to target and the other from target to source, and a cycle consistency loss to keep the similarity of original and cycle-transferred images. SPGAN is designed for object re-ID, which is realised based on CycleGAN, while a similarity-preserving loss is developed to preserve the cross-domain similarities. In our experiments, the settings from (Deng et al., 2018a) and (Yao et al., 2020) are used for person and vehicle domain adaptations, respectively.

Both content-level and pixel-level DA methods provide us with images that have a lower domain gap with the target domain. We then employ important baseline approaches for each task to evaluate the DA method, including ID-discriminative embedding (IDE) (Zheng et al., 2016), the part-based convolution baseline (PCB) (Sun et al., 2018) and Multi-Scale Interaction Network (MSINet) (Gu et al., 2023). When applying PCB in vehicle re-ID, images are divided into vertical stripes.

**Pseudo-label based methods** assign labels to the unlabeled target data based on an encoder. Then, through an iterative process, the labels will be refined, and the encoder updated. Here, the encoder is usually pre-trained on the source domain. Specifically, for object re-ID, PUL (Fan et al., 2018), ECN (Zhong et al., 2019), UDA (Song et al., 2020a) and the recent-state-of-the-art MMT (Ge et al., 2020) are chosen to be evaluated. We use OpenUnReID[4] to implement UDA and MMT. For ECN, we train StarGAN (Choi et al., 2018) to generate the CamStyle for AlicePerson and run the experiments through the code released by ECN authors. PUL uses the re-implemented code. For vehicle re-ID, the image size is set to $256 \times 256$.

## 6.2 EFFECTIVENESS OF CONTENT-LEVEL DA METHOD

**Attribute descent is found to generally improve accuracy.** As an example, the mAP improvement in person re-ID is shown in Fig. 3 A. When attribute descent is used together with other methods, such as IDE, CycleGAN, SPGAN, PUL, ECN, UDA and MMT, it creates an improvement in mAP of +0.19%, +2.04%, +1.34%, 0.52%, +1.92% +1.52% and +0.60%, respectively, under the setting "PersonX→AlicePerson". When Market-1501 is used as target sets, attribute descent also brings obvious improvements. On the AliceVehicle and VeRi-776 datasets, the use of attribute descent helps to create a better source dataset to improve vehicle re-ID results (Fig. 3 B). The mAP improvement on IDE, CycleGAN, SPGAN, PUL, UDA and MMT is +0.89%, +2.34%, +1.39%, +2.61%, +2.57% and +0.11%, respectively for "VehicleX→AliceVehicle".

**Combining attribute descent with pixel DA are more effective than using attribute descent alone.** As shown in Fig. 3 A, for "PersonX→AlicePerson", when combined with CycleGAN, attribute

---

[4]https://github.com/open-mmlab/OpenUnReID

Table 3: Comparison of Methods on AlicePerson and Market-1501 datasets. We show results of (1) direct transfer, (2) content DA, (3) pixel DA and (4) pseudo labels. Except for content DA, all the other methods use PersonX with random attributes as the source domain. ↑mAP means the improvement of each method compared with the baseline IDE. rank-1 (R1) accuracy (%), rank-5 (R5) accuracy (%), rank-10 (R10) accuracy (%), and mAP (%) are reported.

| | Method | PersonX→AlicePerson | | | | | PersonX→Market-1501 | | | | |
| | | R1 | R5 | R10 | mAP | ↑mAP | R1 | R5 | R10 | mAP | ↑mAP |
|---|---|---|---|---|---|---|---|---|---|---|---|
| (1) | IDE | 15.16 | 26.50 | 33.29 | 6.22 | 0 | 36.22 | 51.99 | 59.56 | 14.88 | 0 |
| | PCB | 15.03 | 27.62 | 34.61 | 6.29 | 0.07 | 37.29 | 54.66 | 62.20 | 15.29 | 0.41 |
| | MSINet | 16.41 | 32.17 | 40.94 | 7.27 | 1.05 | 33.40 | 51.66 | 59.65 | 14.34 | -0.54 |
| (2) | Attr. desc. IDE | 16.08 | 27.42 | 32.82 | 6.41 | 0.19 | 38.98 | 54.81 | 62.17 | 16.79 | 1.91 |
| (3) | CycleGAN | 22.54 | 38.10 | 45.29 | 8.75 | 2.53 | 38.30 | 56.74 | 64.10 | 16.12 | 1.24 |
| | SPGAN | 25.84 | 42.25 | 51.55 | 10.24 | 4.02 | 44.80 | 64.04 | 71.29 | 19.30 | 4.44 |
| (4) | PUL | 20.50 | 33.75 | 42.19 | 7.80 | 1.58 | 68.76 | 85.72 | 90.53 | 46.97 | 32.09 |
| | ECN | 37.24 | 56.16 | 63.61 | 14.36 | **8.14** | 71.23 | 85.96 | 94.21 | 43.26 | 28.38 |
| | UDA | 20.76 | 33.03 | 38.96 | 7.44 | 1.22 | 68.50 | 82.93 | 88.45 | 43.18 | 28.30 |
| | MMT | 21.16 | 35.60 | 43.05 | 8.89 | 2.67 | 91.83 | 97.30 | 98.37 | 79.46 | **64.58** |

descent leads to a +2.04% mAP improvement over using CycleGAN alone. In comparison, attribute descent offers an improvement of +0.19% mAP on the IDE baseline. The same trend can be observed when using Market-1501 as the target set. Similarly, for vehicle re-ID, the effect of attribute descent also becomes more prominent when used together with pixel DA. For example, attribute descent combined with SPGAN yields a +2.39% mAP improvement over SPGAN itself (Fig. 3 B), and using attribute descent alone offers an improvement of +0.89% on the IDE baseline mAP score. A probable reason is that synthetic data is too simple to train a robust model for complex real-world tasks. Style transfer increases the complexity of synthetic data by adapting image appearances and introducing noise, which is beneficial for training a more robust model. In this case, the advantages of content DA are more obviously apparent on the target task.

**Attribute descent is less effective when used alone or with pseudo-label methods.** As shown in Fig. 3 A and B, attribute descent offers +0.19% and +0.89% mAP improvements over the IDE baseline on AlicePerson and AliceVehicle, respectively. Compared to the same IDE baseline, the use of attribute descent improves mAP by +1.91%, and +1.02% on Market-1501, and VeRi-776, respectively. The benefits of attribute descent are stable but not significant, especially when compared to the improvements resulting from the combination of attribute descent and pixel DA. The reason could be, object re-ID works using small bounding boxes for learning, which do not contain much of the environmental context. Therefore, the use of content adaptation *only* results in relatively small improvements. However, as previously mentioned, combining content DA with pixel DA yields higher accuracy, indicating that the joint application of these techniques leads to more substantial improvements. For pseudo-label methods, a reason may exist in addition to the previous one. Pseudo-label methods like PUL or MMT usually use an encoder pre-trained on the source to compute pseudo labels. Essentially, these methods are not directly trained with source data. While content DA improves the quality of the source data, the improvement is not significant enough to affect the source pre-trained model. In other words, pseudo-label methods are less sensitive to the quality of source data: without content adaptation, the pre-trained encoder would still be good enough to mine high-quality pseudo labels. Similar observations have been made in existing work (Ge et al., 2020).

## 6.3 EFFECTIVENESS OF TRADITIONAL DA METHOD

**Pseudo-label methods are less effective on AlicePerson and AliceVehicle than on existing target datasets like Market-1501.** In Table 3, pseudo-label based methods yield a significant improvement in accuracy for "PersonX→Market-1501". For example, the mAP score of MMT is 79.46% under this setting, which is consistent with what has been reported in the literature. In comparison, content DA (an mAP score of 16.79%) and pixel DA (an mAP score of 19.30% mAP) are far less effective than pseudo-label methods. However, for "PersonX→AlicePerson", MMT only obtains an mAP of 8.89%, which is lower than the pixel-level SPGAN (an mAP score of 10.24%). The contrasting results are attributed to the data structure of the target datasets. In fact, the training sets in Market-1501 and VeRi-776 are originally intended for supervised learning, where each identity possesses a similar number of samples distributed evenly across cameras. Such an underlying data structure is friendly to

Table 4: Method evaluation on AliceVehicle and VeRi-776. We use VehicleX as the source domain. Similar to the previous table, we show results of (1) direct transfer, (2) content DA, (3) pixel DA and (4) pseudo labels. Notations and evaluation metrics are the same as those in the previous table.

| | Method | VehicleX→AliceVehicle | | | | VehicleX→VeRi-776 | | | |
|---|---|---|---|---|---|---|---|---|---|
| | | R1 | R5 | mAP | ↑mAP | R1 | R5 | mAP | ↑mAP |
| (1) | IDE | 19.63 | 34.51 | 9.08 | 0 | 29.92 | 46.36 | 11.13 | 0 |
| | PCB | 18.85 | 32.18 | 8.89 | -0.10 | 28.61 | 42.37 | 11.36 | 0.23 |
| | MSINet | 25.37 | 38.84 | 10.33 | 1.25 | 40.41 | 54.05 | 15.81 | 4.68 |
| (2) | Attr. desc. IDE | 22.11 | 37.49 | 9.97 | 0.89 | 30.44 | 44.40 | 12.15 | 1.02 |
| (3) | CycleGAN | 25.58 | 44.37 | 11.85 | 2.77 | 42.49 | 60.13 | 17.53 | 6.40 |
| | SPGAN | 29.41 | 49.04 | 13.83 | 4.75 | 46.96 | 63.11 | 19.10 | 7.97 |
| (4) | PUL | 37.77 | 50.74 | 15.73 | 6.65 | 59.77 | 68.47 | 23.00 | 11.87 |
| | UDA | 36.22 | 47.41 | 15.33 | 6.25 | 61.20 | 71.33 | 23.33 | 12.20 |
| | MMT | 60.81 | 71.01 | 27.45 | 18.37 | 80.36 | 86.29 | 36.82 | 25.69 |

clustering-like algorithms such as pseudo-label mining. In comparison, the training set of AlicePerson is much more randomly distributed. For example, many identities have very few examples, and there is no guarantee that images of each identity span multiple cameras. As a result, this poses new challenges for data clustering in pseudo-label methods.

For Vehicle re-ID, there is no huge difference between the results of mining pseudo labels on the VeRi-776 and AliceVehicle datasets (Table 4), but the differences in improvements (↑mAP) are obvious. PUL enhances the mAP score on AliceVehicle by 6.65%, which is approximately half of the 11.87% observed on VeRi-776. MMT improves +25.69% and +18.37% in mAP on VeRi-776 and AliceVehicle, respectively. The improvement magnitude decreases from AliceVehicle to VeRi-776, showing that the settings of AliceVehicle are more challenging for pseudo-label methods than those of VeRi-776. We suggest that the differences in performance on the person re-ID dataset are sharper than on vehicle re-ID dataset because changes in the appearances of cars are limited compared to the changes in people. Despite the randomness of the training set in AliceVehicle, the clustering will not be much weaker than that of the VeRi-776, so there is no huge difference in the final results.

## 6.4 VISUALIZATION OF DA METHODS

To show the effect of content and pixel DA, as has been a focus of discussion above, here we provide visualization examples of these two methods. For object re-ID, Fig. 4 shows image examples before and after the content/pixel adaptation has taken effect. Specifically. we have the following observations. Firstly, attribute descent allows for key attributes of the synthetic data to tend toward the target domain. For example, the cameras of the target domain are mostly of the same height as people, and attribute descent can effectively adjust source cameras to this angle. In vehicles, the target images are mostly in side view, and accordingly, the edited source images exhibit a similar trend. Secondly, SPGAN causes obvious changes bringing synthetic images closer to the target style. For example, the skin color of synthetic persons and the illumination/tone of vehicles becomes more realistic and closer to the target. Thirdly, when content and style alignment both exist, the source data exhibit the highest similarities to the target domain, which explains the superior performance of attribute descent in conjunction with pixel DA in Fig. 3 A and B.

## 7 DISCUSSION AND FUTURE WORK

This section will further discuss some open questions, challenges and interesting research problems for future investigations. More discussions can also be found in the Appendix.

**Benefits of learning from synthetic data.** Major advantages of synthetic data include the following. The first is *low cost*. When annotating AlicePerson, it takes three people 8-9 hours each to annotate 80 IDs across 5 cameras. In comparison, after crafting and modifying 3D person models, generating synthetic training data only takes minutes. Second, labels of synthetic data are perfectly *accurate*.

In a less exploited domain, synthetic data is *controllable* and *customizable*. For example, many tasks (*e.g.*, semantic segmentation and object detection) require an agent to be robust in various scenes under low lighting or extreme weather. Such data is usually difficult to acquire in the real world, but

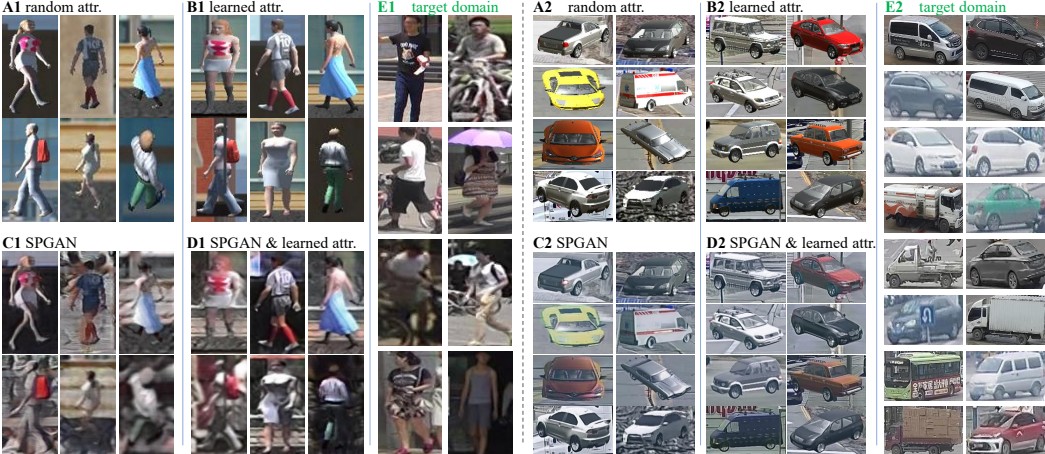

Figure 4: Sample images on person (**left**) and vehicle re-ID (**right**) before and after content/pixel domain adaptation. There are source images generated by **(A1-2)** random attributes, **(B1-2)** attribute descent, **(C1-2)** SPGAN and **(D1-2)** SPGAN & attribute descent. **E1-2:** show samples from the target domain. We find that attribute descent changes the viewpoint and illuminations *etc*., visual factors, of the objects, while SPGAN increases image blurring and adapts the image color to the target style.

can be easily generated through graphic engines. Moreover, synthetic data allows us to evaluate the robustness of algorithms by setting up various test sets with changing visual factors. In addition, synthetic data can alleviate concerns about data privacy and ethics.

**Challenges of organizing synthetic datasets and improvement directions.** Obtaining a variety of 3D object/scene models is the key to the successful use of synthetic data. The ideal goal is the development of a virtual 3D world from which we can generate various synthetic data for a variety of computer vision tasks. However, the challenge is the lack of synthetic assets, such as 3D objects and texture materials. Besides manually crafting 3D models, it would be beneficial to resort to 3D vision-related research, *e.g.,* automatically generating 3D models based on real-world images (Wu et al., 2020), and automatically changing the appearance of 3D models to acquire diverse data. Once the 3D models of a synthetic engine are ready, it will be fast and inexpensive to generate as much data as needed. In the future, we will gradually expand the Alice benchmarks to include more tasks and provide more complex and higher-quality synthetic data.

**Generalization of conclusions drawn based on the Alice benchmarks.** In most cases, we believe that conclusions drawn based on the Alice benchmarks are generalizable to other target datasets as well. In Table 3, and 4, in addition to the Alice targets, we also use existing datasets as the target domain and report the results of different methods. It is evident that there are similar trends of results between Alice and other existing datasets. For example, SPGAN (Deng et al., 2018a) outperforms CycleGAN, and MMT (Ge et al., 2020) obtains higher results than UDA (Song et al., 2020a), both of which are consistent with reports in the literature. An exception is a comparison between pseudo-label methods and pixel DA methods: the former is inferior for AlicePerson, which contradicts prior expectations. We have discussed this point in Section 6.3.

## 8 CONCLUSION

This paper introduces the Alice benchmarks, including a series of datasets and an online evaluation server to facilitate research on "synthetic to real" domain adaptation. Our current version includes two tasks: person and vehicle re-ID. For each task, fully editable synthetic data and newly collected real-world data are used as the source and target respectively. We provide insightful evaluation and discussion of some commonly used DA methods on the content and pixel level, as well as some pseudo-label methods. We have discussed some interesting future research problems that are enabled by this platform, which may bring new and exciting ideas to the community. The Alice project is our long-term project. In our future work, we will try to include more tasks beyond the existing re-ID task by welcoming open-source collaboration, with the aim of covering more objects in the visual world.

ACKNOWLEDGEMENT

We thank all anonymous reviewers and AC for their insightful comments and suggestions in improving this paper. This research was funded in part by the ARC Discovery Project (DP210102801) to Liang Zheng, and the ARC Discovery Grant (DP220100800) to Hongdong Li.

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

APPENDIX

In the Appendix, we present the summary of existing synthetic datasets (Section A), the dataset annotation procedure (Section B), sources of datasets and DA method (Section C and Section D), as well as discussion and future work (Section E).

## A  SUMMARY OF EXISTING SYNTHETIC DATASETS

Table 5 presents some synthetic datasets published in recent years, designed for various purposes. This indicates synthetic data has emerged as a powerful tool, offering flexibility in data design and addressing challenges related to data scarcity and privacy concerns. In line with this, our research endeavors to push the boundaries of synthetic data in the field of object re-identification.

Table 5: Example synthetic datasets released in recent years.

| name | task | data format | engine |
|------|------|-------------|--------|
| SOMAset (Barbosa et al., 2018) | person re-ID | image | Unreal |
| SyRI (Bak et al., 2018) | person re-ID | image | Blender |
| PersonX (Sun & Zheng, 2019) | person re-ID | image/model | Unity |
| RandPerson (Wang et al., 2020) | person re-ID | image/video | Unity |
| UnrealPerson (Zhang et al., 2021) | person re-ID | image | Unreal |
| ClonedPerson (Wang et al., 2022) | person re-ID | image/model | Unity |
| WePerson (Li et al., 2021) | person re-ID | image/model | Script Hook V |
| GPR (Xiang et al., 2020) | person re-ID | image | GTA V |
| VehicleX (Yao et al., 2020) | vehicle re-ID | image/model | Unity |
| PAMTRI (Tang et al., 2019) | vehicle re-ID | image | Unreal |
| SYNTHIA (Ros et al., 2016) | segmentation | image | Unity |
| GTA5 (Richter et al., 2016) | segmentation | image | GTA V |
| SceneX (Xue et al., 2021) | segmentation | image/model | Unity |
| SAIL-VOS (Hu et al., 2019) | segmentation | image/video | GTA V |
| GCC (Wang et al., 2019b) | crow counting | image | GTA V |
| G2D (Doan et al., 2018) | SfM | image/engine | GTA V |
| SDR (Prakash et al., 2019) | car detection | image/model | Unreal |
| CARLA (Dosovitskiy et al., 2017) | depth estimation & segmentation | image/model | Unreal |
| Sim4CV (Mueller et al., 2017) | navigation & tracking | image/model | Unreal |
| Virtual KITTI (Gaidon et al., 2016) | object tracking & detection & segmentation | image/video | Unity |
| synthetic2real (Peng et al., 2018) | classification & detection | image | CAD |
| VisDA (Peng et al., 2017) | classification & segmentation | image | CAD & GTA V |
| AI2-THOR (Kolve et al., 2017) | navigation & | image & model | Unity |
| RoboTHOR (Deitke et al., 2020) | navigation & tracking | image | Unity |
| Meta-Sim (Kar et al., 2019) | car detection | image | Unreal |
| SVIRO (Cruz et al., 2020) | interior vehicle & sensing | image | Blender |
| MultiviewX (Hou et al., 2020) | multi-view & detection | image | Unity |

## B  DATA ANNOTATION

Figure 5 displays the GUI, where **Top** and **Bottom** are examples for annotating different datasets. Annotators collaborate based on the video recording timestamps. Each annotator oversees 1-2 cameras, collectively identifying individuals with the same ID. After annotation, we conducted multiple rounds of checks on the data to ensure the accuracy of annotation. Our data is annotated

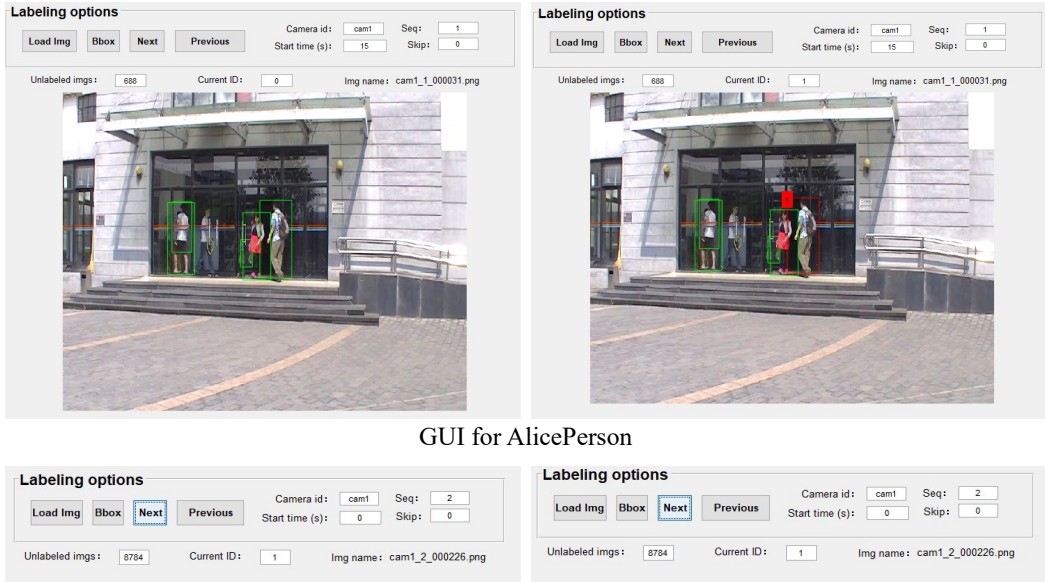

GUI for AlicePerson

GUI for AliceVehicle

Figure 5: **Graphical user interface (GUI) used in our annotation process.** AlicePerson (Top) and AliceVehicle (Bottom). Annotators work together according to the time of video recording. Each annotator is in charge of 1-2 cameras, looking for the same ID together.

by a data annotation company, which has a legitimate business license and meets the required wage levels (24 AUD / Hour) of the government.

Test set statistics are shown in Fig. 6. For AlicePerson, we can observe that the 4th camera captures a larger number of unique IDs compared to the other cameras from **A** of Fig. 6. Additionally, it is noticeable that most IDs appear in 4 or 3 cameras. For AliceVehicle, each camera includes about 300-800 images, and most IDs appear in 4 - 6 cameras. In the test set, we do not have ID only appears in one camera except for person on distractor images.

We would like to provide more discussion of our datasets. Beyond the clusterability, our real-world datasets exhibit several significant differences when compared to existing datasets like Market-1501: 1) Target training data is more consistent with real-world distributions because they are not manually selected by humans. 2) AlicePerson and AliceVehicle are more challenging datasets because existing methods get lower improvements on them than on existing datasets. 3) The test labels of AlicePerson and AliceVehicle are hidden, which will offer a unified, fair, and challenging benchmark.

Meanwhile, datasets are only part of Alice Benchmarks. Alice provides: 1) an online evaluation platform which as far as we know is the first online evaluation server for person/vehicle re-identification; 2) evaluations and analysis of different domain adaptation methods, which can provide some insights into learning from synthetic data; 3) discussions of interesting questions and future directions.

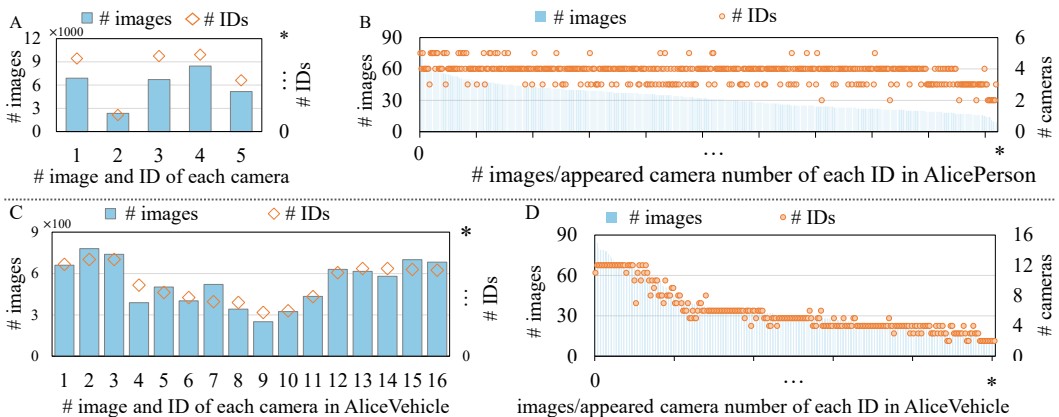

Figure 6: Statistics of the AlicePerson/AliceVehicle test sets. **A** and **C**: For each camera of AlicePerson and AliceVehicle, we display the number of images and object IDs that have appeared in that camera. **C** and **D**: For each ID of AlicePerson and AliceVehicle, we show the number of images and how many cameras the ID appeared. Note that the number of IDs in the test set is hidden for online evaluation. # is the notation of "the number of".

## C   LINKS TO DATASETS

We make the real data we collected for Alice benchmarks publicly available on the links below:

**Alice-train**: `https://drive.google.com/file/d/19sQdxFwF9LTmK8BjhINjWvp8o-UwjnRc/view?usp=sharing`

**Alice-validation-test**: `https://drive.google.com/file/d/1SAlSwfxUZ0QQCkja5BuBbVk6x0gOnkSh/view?usp=drive_link`

The synthetic data we reused is also available on these links:

**PersonX**: `https://github.com/sxzrt/Instructions-of-the-PersonX-dataset`

**VehicleX**: `https://github.com/yorkeyao/VehicleX`

## D   SOURCE CODE OF DA METHODS

**Attr. desc.**: `https://github.com/yorkeyao/VehicleX`

**IDE**: `https://github.com/layumi/Person_reID_baseline_pytorch`

**PCB**: `https://github.com/layumi/Person_reID_baseline_pytorch`

**CycleGAN and SPGAN**: `https://github.com/Simon4Yan/eSPGAN`

**PUL**: `https://github.com/hehefan/Unsupervised-Person-Re-identification-Clustering-and-Fine-tuning`

**ECN**: `https://github.com/zhunzhong07/ECN`

**UDA**: `https://github.com/open-mmlab/OpenUnReID`

**MMT**: `https://github.com/open-mmlab/OpenUnReID`

## E   DISCUSSION AND FUTURE WORK

**Understanding the domain gap between the synthetic and the real.** The domain gap is usually studied with real-world datasets. For example, Torralba *et al.* (Torralba & Efros, 2011) analyze and summarize different types of domain bias, such as the selection bias (different collection sites, websites, *etc*) and the caption bias (different viewpoints, resolutions *etc*). In the context of "synthetic to real", the domain gap may be different from that in "real to real" DA. For example, the difference between 3D object models and real-world objects may create a domain gap that does not occur under the "real to real" setting. It is still largely unknown whether existing "real to real" DA methods

Table 6: FID values between target and synthetic data with (Attr. desc) and without (Random) content-level DA. A lower FID value means a smaller distance between synthetic and real data.

| Target datasets | Random | Attr. desc |
|---|---|---|
| Market-1501 | 104.78 | 83.24 |
| AlicePerson | 134.00 | 121.75 |
| VeRi-776 | 80.22 | 57.45 |
| AliceVehicle | 78.05 | 56.07 |

can handle such new problems. Also, simulated data has lower diversity than real-world data in complex environments. However, given that synthetic data can be simulated in large amounts, would it compensate for its lower data diversity? In this regard, an interesting question would be whether synthetic data makes an inferior source domain to real data. In other words, given the same target domain, which should we choose as a source: real data or editable synthetic data?

**Designing "synthetic to synthetic" DA evaluation protocols for comprehensive evaluation of DA methods, dissecting the domain gap and providing references for "real to real" DA.** The evaluation and understanding of DA methods are limited by the scale and variation of both the source and the target data. This problem can be overcome by controllable and customizable synthetic data. For example, we can generate source and target data with similar styles but very different content distributions for semantic segmentation. Conversely, we can synthesize data with similar content distributions but very different styles. With various data settings, we can conduct well-directed and comprehensive evaluations of domain adaptation methods, which will also be helpful for understanding the domain gap. On the other hand, although quantitatively defining the domain gap caused by changes in various visual factors is rather infeasible, it is possible to analyze the domain gap in a higher capacity by using synthetic data. For example, the illumination of source data can be gradually changed for a fixed target set to analyze the influence of illumination differences on domain adaptation. Similarly, we can study the relative changes of various visual factors between the source and target domain, which may bring us a better and deeper understanding of the domain gap.

**Need for new pseudo-label methods for object re-ID under more realistic settings.** The object re-ID results in Tables 3 and 4 show that state-of-the-art pseudo-label methods, such as MMT, demonstrate high performance when using the Market-1501 and VeRi-776 as target sets, but do not work well on the Alice benchmarks (refer Section 6.3). Given the decreased performance of pseudo-label methods, we should rethink the clustering strategies used for more realistic target sets. It would be interesting to study how to better leverage the camera style of the target set to assist with data clustering. Adopting this strategy, ECN yields the highest results for AlicePerson (Table 3).

**New strategies to effectively use synthetic data to bridge the gap with the real world.** This paper discusses several existing strategies such as pixel DA and the relatively new content DA. For the latter, there are still many open questions, *e.g.,* the design of new similarity measurements for distributions apart from the FID score used in (Yao et al., 2020; Sun et al., 2021) (Table 6 shows some results of the FID values). Moreover, the characteristics of a good training set are still largely unknown. In this regard, having a smaller domain gap between training and testing sets might not be the only objective. Other indicators such as appearance diversity or even noise are worth investigating. Furthermore, a variety of annotation types can be obtained from synthetic data, such as pixel-level semantics, bounding boxes and depth. These annotations may facilitate research in multi-task learning, which would also potentially improve real-world performance. In addition, it is an interesting task to explore how to combine simulation 3D models and real images in learning.

**Is it possible to evaluate the quality of generated data and its effect on enhancing results?** Yes, but it is a complex issue. In this paper, we offer some preliminary discussions. The assessment of synthetic data quality includes two primary dimensions: annotation quality and image quality. In terms of annotation quality, synthetic data have near-perfect label accuracy, largely thanks to an automated annotation process. This involves creating images with annotations derived from the IDs of 3D models of people and vehicles. Once these models are set up, a system automatically generates annotations, significantly reducing the chance of labeling errors. Consequently, the reliability of labels in our synthetic data is very high. Image quality, on the other hand, is affected by various factors, such as resolution and the sophistication of the 3D models used. Sun & Zheng (2019) have shown in Figure 4 in their paper that utilizing high-resolution images leads to improved outcomes. Furthermore, while Man & Chahl (2022) explored the impact of using different graphics engines

like Unity and Unreal on data quality, they conclude that the choice of the engine plays a lesser role compared to the quality of the 3D models themselves. Therefore, we believe that the generation of high-quality synthetic data is predominantly influenced by the quality of 3D assets over the choice of graphics engine. Despite the discussion above, understanding how the quality of synthetic data affects the results presents a highly complex question and warrants a separate investigation.

**Can we finally get rid of real-world data when proposing new models?** While state-of-the-art techniques in computer vision have been developed on real-world datasets, it would be interesting to study whether we can resort to pure synthetic data (training + testing) in model development, underpinned by the increasing ethics concerns in artificial intelligence (AI). Several challenges are yet to be resolved. Firstly, diverse, complex, and realistic synthetic data needs to be created. Secondly, it is necessary to confirm that training and testing on synthetic data are analogous to real-world performance. Thirdly, we need to explore the optimum composition of test data so as to comprehensively evaluate model performance. Apart from these challenges, there are many task-specific problems to be considered. At this point, it has not been determined how real-world test data can be replaced with synthetic data, but we will consider this to be an option with future higher-fidelity data generators and stronger theoretical support.

**What are the potential limitations of using synthetic data to enhance learning and inference in real-world scenarios?** Beyond the commonly acknowledged domain gap between synthetic and real-world data, a significant challenge in leveraging synthetic data lies in the complexity of creating specific datasets for targeted tasks, like those in medical imaging. Nonetheless, there is a promising movement within the research community towards generating more varied synthetic datasets that encompass a wider array of objects. An illustrative example is the Infinigen dataset (Raistrick et al., 2023a), demonstrating the growing capabilities of synthetic data generation. Therefore, even with these limitations, the scope for using synthetic data is broadening. We expect its application across an increasing number of scenarios in the future.

**What is the relationship between the control of synthetic scene creator and its influence on the content gap with real-world data?** Initially, human intervention in adjusting the content of simulated scenes can effectively minimize the content gap by exerting control over various factors, such as lighting conditions, object placement, and background variations. However, there are significant limitations, including the high cost associated with generating large-scale datasets.This constraint has significantly motivated the research community to explore machine learning methods aimed at automating the reduction of content gaps. Furthermore, content-level DA operates in a manner akin to manual control of visual attributes within scenes, although it remains far from perfect. This approach employs the Fréchet Inception Distance (FID) to evaluate the quality of synthetic data and subsequently adjusts the parameters governing visual attributes in the simulation environment. This iterative process aims to progressively diminish the content gap and enhance alignment between synthetic and real data distributions.

The experiments were conducted on a server equipped with four RTX-2080TI GPUs and a 16-core AMD Threadripper CPU @ 3.5Ghz.

