# OpenReview forum: "Alice Benchmarks: Connecting Real World Re-Identification with the Synthetic"
_ICLR.cc/2024/Conference — ICLR 2024 poster_

### Official Review · Reviewer_6iq6 · 2023-10-13

**Soundness:** 2 fair
**Presentation:** 3 good
**Contribution:** 2 fair
**Rating:** 6
**Confidence:** 5

**Summary:**

The paper focuses on the research of synthetic data for object re-identification. Although using the name of object re-identification, it only focuses on persons and vehicles. The main contribution of the paper is the proposed Alice benchmarks, which include both synthetic and real-world data. Specifically, an application is training a model with fully-labeled synthetic data and then tuning it on the unlabeled real-world data. A significance of the paper is its real-world data do not have pre-define “cluster” architecture, which is more practical. The authors try to regard it as a successor
 of the famous Market benchmark.

**Strengths:**

1. Focusing on synthetic data is good; which does not invade privacy;
2. Good writing and easy to follow;
3. Some insightful discussions in DISCUSSION AND FUTURE WORK

**Weaknesses:**

Although with these strengths, I think the paper is still far away from the acceptance of ICLR 2024.
1. The contribution is not enough. I fully understand the importance of the benchmark. But as far as I am concerned, the paper’s biggest significance is the “un-clustered” data architecture. Some other contributions as benchmarking existing works and discussion. They cannot be considered as significance due to (1) no significance contribution beyond other works. (2) no novel method is proposed to customize the new-labeled synthetic to real dataset.
2. The domain adaptation seems not suitable for synthetic to real datasets. The gain of synthetic data is easy and straightforward. Therefore, we should generate more diversed synthetic data to achieve domain generalization rather than domain adaptation. Please refer to ClonePerson and RandPerson for this kind of research;
3. The authors discuss the legitimate in data-labeling, however, the main step that incurs the privacy problem is capturing data. How can your data captured in Figure 2 protects the privacy of persons and vehicles. As far as I concern, the passerbys cannot know their cars or themselves are trained for AI models. A license from local government is not enough, it seems to be the human right matters;
4. The authors try to provide some understanding, however, these understandings are not well-defined by mathematics. Instead, they are some intuitive ones without proof. Some understanding in theory is expected.

**Questions:**

Please see the weakness.,

**Details Of Ethics Concerns:**

The authors discuss the legitimate in data-labeling, however, the main step that incurs the privacy problem is capturing data. How can your data captured in Figure 2 protects the privacy of persons and vehicles. As far as I concern, the passerbys cannot know their cars or themselves are trained for AI models. A license from local government is not enough, it seems to be the human right matters;

---

> ### Author Response · Authors · 2023-11-17
> **Response to Reviewer 6iq6: part-1**
>
> We thank the reviewer for many insightful comments. We answer the questions in what follows. Please let us know if further clarification is needed.
>
> **Q1: The contribution is not enough.**
>
> Thank you for your comments. We aim to clarify the contributions of this work to provide a clearer understanding:
>
> First, the Alice benchmarks include an online service for evaluating domain adaptation algorithms in object re-identification, which as far as we know is the first online evaluation server in the re-ID community. This service specifically focuses on learning from synthetic data and testing on real-world data.
>
> Second, two challenging real-world datasets are collected for person and vehicle re-ID evaluation when learning from synthetic data. Note that the "un-clustered" data architecture is not their only key characteristic. Additionally, they offer unique features: 1) The target training data more accurately reflect real-world distributions as they are not manually selected. 2) Both AlicePerson and AliceVehicle present more challenging test scenarios compared to existing datasets. 3) The test labels in these datasets are hidden, providing a unified, fair, and challenging benchmarking environment.
>
> Third, We conduct evaluations of some existing common domain adaptation methods on our two tasks. Results and analysis provide insights into the characteristics of synthetic and real-world data.
>
> Fourth, our work introduces new research problems enabled by Alice, particularly in understanding the "synthetic to real" domain adaptation challenge.
>
> Compared to existing works, our paper systematically evaluates and discusses various domain adaptation methods for learning from synthetic data. As far as we know, there is no previous work conducting this kind of evaluation.  Therefore, we respectfully disagree with the comment that  ''no significant contribution beyond other works".
>
> Regarding the  ''no novel method being proposed to customize the new-labeled synthetic to real dataset", this paper provides a friendly editing interface where researchers can generate their own source dataset with an arbitrary number of images under **manually controllable environments**. This interface, combined with our discussions on **content-level domain adaptation methods**, lays the groundwork for future explorations in customizing synthetic data. We will study this in our future work.
>
> ---
> **Q2: The domain adaptation seems not suitable for synthetic to real datasets. The gain of synthetic data is easy and straightforward. Therefore, we should generate more diverse synthetic data to achieve domain generalization rather than domain adaptation. Please refer to ClonePerson and RandPerson for this kind of research.**
>
> Thank you for your insightful comments. It is important to recognize that domain adaptation and domain generalization are distinct approaches, each with its unique applications, particularly when utilizing synthetic data for training. While it is challenging to definitively determine which approach is more suitable for synthetic to real datasets, studying the potential of domain generalization is a good idea.
>
> In our current work with Alice, we have focused on domain adaptation, partly because acquiring some unlabeled target data is relatively low-cost, making it a feasible starting point. However, we acknowledge the potential and relevance of domain generalization in this context, as the reviewer suggested. We will extend the scope of Alice to include domain generalization in our future work. This expansion will enable us to investigate the potential effectiveness of synthetic data across both domain adaptation and generalization contexts.
>
> ---
> **Q3: Privacy of persons and vehicles: How the data captured in Figure 2 protects the privacy of people and vehicles.**
>
> Thank you for highlighting the critical issue of privacy. As we have noted on page 2 (bottom) of our paper, we have taken specific measures to protect the privacy of individuals in Alice. This includes manually blurring human faces and vehicle license plates to prevent identification. Additionally, the dataset we released consists solely of bounding boxes for persons and vehicles, rather than full images like in Figure 2. Moreover, our dataset, referred to as Alice in the paper, is distributed under the Creative Commons Attribution-NonCommercial 4.0 International License (CC BY-NC 4.0). This license restricts the use of Alice for non-commercial purposes, further mitigating potential privacy concerns.
>
> We also recognize the value of using synthetic data as an effective strategy to avoid privacy issues. In this regard, Alice is intended to encourage and facilitate research in this direction. By adopting these measures, we have minimized privacy risks as much as possible while contributing to the advancement of research in the field.

---

> ### Author Response · Authors · 2023-11-17
> **Response to Reviewer 6iq6: part-2**
>
> **Q4: Understandings provided in the paper are not well-defined by mathematics. Instead, they are some intuitive ones without proof. Some understanding in theory is expected.**
>
> Thank you for your insightful comments. We wish to clarify that the primary focus of this paper is on establishing a benchmark rather than developing theoretical foundations. Given the nature of current tasks, *i.e.*, Person and Vehicle re-identification, in Alice, the emphasis in existing literature has largely been on empirical approaches, with less focus on theoretical proof. This background has shaped our paper, which leans towards experimental illustrations.
>
> Meanwhile, we recognize the importance of supporting our understanding with more evidence and will add additional visualizations in the revised paper to support the intuitive understanding we present. These visualizations will aim to provide a more straightforward representation of our understanding.
>
> Additionally, in response to the need for theoretical grounding, we will enrich the paper with relevant theoretical discussions. For instance, we will discuss the impact of domain adaptation methods on domain gaps based on FID values that can indicate domain gaps. From a theoretical standpoint, synthetic data with a small FID could be aligned well with target test data, so it can be used to train a more effective model for the test scenarios. Such theoretical insights will complement our experimental observations, providing a better explanation of the understandings in the paper.

---

> > ### Comment · Reviewer_6iq6 · 2023-11-17
> > **Happy to rise scores**
> >
> > After checking the rebuttals, most of my concerns are solved. Therefore, I decide to raise the score to 6. Happy to see this work accepted.

---

### Official Review · Reviewer_ZhRK · 2023-10-31

**Soundness:** 2 fair
**Presentation:** 3 good
**Contribution:** 1 poor
**Rating:** 6
**Confidence:** 5

**Summary:**

This paper proposes a new benchmark for synthetic-to-real object reid. The new benchmark involves two real-world data, i.e., AlicePerson and AliceVehicle, and provides some analysis on them.

**Strengths:**

The strong clusterability in current data actually brings a risk to build a robust UDA reid method, which deserves atttention.

**Weaknesses:**

1. The synthetic data in benchmark are from existing datasets, i.e., PersonX and VehicleX, while the real-world data (AlicePerson and Alice Vehicle) in benchmark shows less advantage compared to current off-the-shelf reid datasets. The strong clusterability issue is the only difference in my opinion. However, for AlicePerson, I do NOT think adding distractor and junk is a good way to solve the strong clusterability issue compared. For AliceVehicle, in Table.4, I do NOT find much difference between AliceVehicle and VeRi776 on the strong clusterability issue.
2. The UDA methods compared in this benchmark (Fig.3&Tab.3&Tab.4) are out of fashion, new methods should be involved.

**Questions:**

Beside the introduce of the strong clusterability issue, I can not find any new things in this benchmark. Please see the weakness for details.

---

> ### Author Response · Authors · 2023-11-17
> **Response to Reviewer ZhRK**
>
> We thank the reviewer for many insightful comments. We answer the questions in what follows. Please let us know if further clarification is needed.
>
> **Q1: The synthetic data in the benchmark are from existing datasets, i.e., PersonX and VehicleX**
>
> Indeed, PersonX and VehicleX serve are existing datasets, but we have significantly modified and integrated them into a new framework within Alice. This integration includes **a user-friendly editing interface**, enabling researchers to create customized source datasets. This interface allows for the generation of a varied number of images under environments that are manually adjustable, offering greater flexibility and control in dataset creation.
>
> **Q2: The real-world data (AlicePerson and AliceVehicle) in benchmark shows less advantage compared to current off-the-shelf re-ID datasets: The strong clusterability issue is the only difference.**
>
> We respectfully disagree with this comment. Beyond the clusterability issue, our real-world datasets exhibit several significant differences when compared to existing datasets like Market-1501:
>
> 1) Target training data is more consistent with real-world distributions because they are not manually selected by humans.
>
> 2) AlicePerson and AliceVehicle are more challenging datasets because existing methods get lower improvements on them than on existing datasets.
>
> 3) The test labels of AlicePerson and AliceVehicle are hidden, which will offer a unified, fair, and challenging benchmark.
>
> Meanwhile, datasets are only part of Alice Benchmarks. Alice provides:
>
> 1) an online evaluation platform which as far as we know is the first online evaluation server for person/vehicle re-identification.
>
> 2) evaluations and analysis of different domain adaptation methods, which can provide some insights into learning from synthetic data.
>
> 3) discussions of interesting questions and future directions.
>
> Furthermore, the Alice project is our long-term project. In our future work, we will try to include more tasks beyond the existing re-ID task by welcoming open-source collaboration, with the aim of covering more objects in the visual world. Therefore, we think the current Alice and its potential version will contribute to the community for studying learning from synthetic data and related topics. We will add these discussions in our revised paper to provide readers with a more complete understanding of the Alice Benchmarks.
>
> **Q3: For AlicePerson, adding distractor and junk is not a good way to solve the strong clusterability issue compared. For AliceVehicle, in Table.4, I do NOT find much difference between AliceVehicle and VeRi776 on the strong clusterability issue.**
>
>
> Thank you for your comments. However, there seems to be a misunderstanding regarding the settings of Alice. Firstly, the inclusion of distractors and junk in AlicePerson is not intended to address clusterability issues. In image retrieval and person re-ID communities, incorporating low-quality bounding boxes, often termed "distractors," into test sets is a standard practice. For instance, Oxford5K has 25\% of distractors in its gallery images, while Market-1501 includes 2,793 distractors in its 19,735 gallery images. The reduced clusterability in Alice target training sets is caused by collecting data directly from real-world scenarios without manual selection.
>
> Regarding the comparison of AliceVehicle and VeRi776, our paper has discussed this in the second paragraph of Section 6.3:
>
> >differences in performance on the person re-ID dataset are sharper than on vehicle re-ID dataset due to the fact that changes in the appearances of cars are limited compared to the changes in people. Despite the randomness of the training set in AliceVehicle, the clustering will not be much weaker than that of the VeRi-776 training set, so there is no large difference in their final results
>
> Numerically, the results may appear similar, but a closer examination reveals notable differences. For instance, when comparing improvements over the baseline IDE, most methods on AliceVehicle achieve about 50\% of the improvement they do on VeRi-776. As an example, PUL's improved mAP score on AliceVehicle is 6.65\%, roughly half of its 11.87\% improvement on VeRi-776. This suggests that AliceVehicle presents more challenging conditions than VeRi-776.
>
> We will add these discussions to our revised paper.
>
> **Q4: The UDA methods compared in this benchmark are out of fashion, new methods should be involved.**
>
> Thanks for your kind suggestion. In our current paper, we have focused on evaluating the most commonly used DA methods to establish a \red{reliable benchmark}. These methods have been frequently cited and utilized in recent re-identification (re-ID) literature. Meanwhile, we also compared different baseline methods in which MSINet (CVPR2023) is a very recent work. Following the suggestion of the reviewer, we will try to add new methods in the revised paper.

---

> > ### Comment · Reviewer_ZhRK · 2023-11-21
> > **Happy to raise score**
> >
> > Thanks for the reviewers' detailed responses. Most of my concerns are solved, so i'm happy to raise my score.

---

### Official Review · Reviewer_txUp · 2023-10-31

**Soundness:** 3 good
**Presentation:** 3 good
**Contribution:** 4 excellent
**Rating:** 8
**Confidence:** 4

**Summary:**

In this work authors present a new benchmark, namely Alice for well-known re-identification of people and Vehicles. Specifically, the dataset is constructed with the domain adaptation in mind. This provides a great opportunity for DA research and more consistent basis for comparison. In the dataset authors tried to ensure that the collected real-world dataset resembles the unconstrained data in the wild. Additionally, the benchmark comes with an online evaluation system for a fair comparison.

**Strengths:**

The paper is well-written, and the motivation is quite clear. The paper has a very nice structure and flow. A unified evaluation framework is gathered to ensure fairness across results. Additionally, authors provided baseline results based on well-knows domain adaptation works and techniques. This provides unique opportunity to enhance domain adaptation research. Moreover, authors showed how established techniques such as unsupervised clustering of images that result in reasonable performance gains in existing benchmarks does not translate to their dataset highlighting the undesired biases that sometime are over-looked.

**Weaknesses:**

Despite not being quite clear at this point, it would be nice if authors could provide some discussion on the potential limit that synthetic data can benefit the learning and inference on real-world data. Specifically whether there are some inherent barriers.

**Questions:**

-

---

> ### Author Response · Authors · 2023-11-17
> **Response to Reviewer txUp**
>
> We thank the reviewer for many insightful comments. We answer the questions in what follows. Please let us know if further clarification is needed.
>
> **Q: It would be nice to provide some discussion on the potential limit that synthetic data can benefit the learning and inference on real-world data. Specifically, whether there are some inherent barriers.**
>
> Thank you for your valuable suggestion. In this paper, we primarily focus on exploring the domain gap between synthetic and real-world data, which is thought as a key barrier in this field. Additionally, in Section 7, we have provided some related discussions of other possible barriers. For instance, under the topic "Challenges of organizing synthetic datasets and improvement directions," we highlight the limitations caused by the lack of synthetic assets. In the section "Generalization of conclusions drawn based on the Alice benchmarks," we discuss the applicability of conclusions derived from synthetic data to real-world scenarios.
>
> Moreover, an additional challenge in using synthetic data is the difficulty of synthesizing data for specific tasks, such as in the field of medical imaging. However, the research community is showing a promising trend towards developing more diverse synthetic datasets including a broader range of objects. For example, the Infinigen dataset presented at CVPR2023 shows the expanding capabilities of synthetic data.
>
> Following your suggestion, we will incorporate these additional insights and discussions into the revised version of our paper, thereby providing a more comprehensive understanding of the potential and limitations of synthetic data in learning and inference.

---

> > ### Comment · Reviewer_txUp · 2023-11-22
> >
> > Thank you for the response and the clarification. Based on all reviewers' comments and the respective responses provided by the authors, I keep my original rating of acceptance.
> >
> > Best,
> > txUp

---

### Official Review · Reviewer_dCYN · 2023-11-02

**Soundness:** 2 fair
**Presentation:** 3 good
**Contribution:** 2 fair
**Rating:** 6
**Confidence:** 4

**Summary:**

This paper presents the Alice benchmarks, a resource designed for advancing research in "synthetic to real" domain adaptation, particularly in the context of person/vehicle re-identification (re-ID). The primary aim is to train models from synthetic data that perform effectively in real-world scenarios. The paper outlines the benchmark settings, analyzes common domain adaptation methods, and sets up an online server for convenient and fair method evaluation. The Alice project is a long-term effort with plans to expand to cover more objects in the visual world through open-source collaboration.

**Strengths:**

1)	This paper offers a good benchmark for "synthetic to real" domain adaptation in the domains of person and vehicle re-identification, significantly enhancing progress in these areas.
2)	Lots of experiments have been constructed, and many interesting findings have been obtained.

**Weaknesses:**

1)	The authors should consider the impact of different synthetic data generation strategies on domain adaptation effects, such as using different graphics engines, different attribute distributions, different noise levels, etc.
2)	The quality of the generated data and labels is not clearly discussed. I recommend measuring the quality of the generated data and the impact of improving quality on the results.
3)	The title of this paper is biased. It focuses on the person/vehicle re-identification, while the title uses object re-identification. The scope of the title is too large. Although the authors claim that more objects will be added later, for the purposes of this article, it is limited to the field of person/vehicle re-identification.

**Questions:**

None

---

> ### Author Response · Authors · 2023-11-17
> **Response to Reviewer dCYN**
>
> We thank the reviewer for many insightful comments. We answer the questions in what follows. Please let us know if further clarification is needed.
>
> **Q1: It should be considered that the impact of different synthetic data generation strategies on domain adaptation effects.**
>
> Thank you for your suggestion. In our paper, we have included some discussions regarding the impact of various synthetic data generation strategies on the effectiveness of domain adaptation.
>
> For example, Section 6 discusses the impact of varying attribute distributions, specifically comparing random attribute selection and attribute descent methods, on domain adaptation results. Our results indicate that re-ID domain adaptation models, when trained on synthetic data refined by attribute descent, show improved performance. Additionally, Table 6 presents the FID scores comparing target real-world data with synthetic data, with and without attribute adjustment. These results reveal that synthetic data with learned attributes have a smaller domain gap with real-world data compared to randomly attributed data.
>
> Furthermore, in Section E, we explore other potential strategies for effectively utilizing synthetic data. For example, it is mentioned that
> > the characteristics of a good training set are still largely unknown. In this regard, having a smaller domain gap between training and testing sets might not be the only objective. Other indicators such as appearance diversity or even noise are worth investigating.
>
> While these possible directions are interesting, it is beyond the scope of a single paper to explore every potential strategy in depth. Therefore, our current work begins by benchmarking common strategies and offering insightful discussions to lay a foundation for future exploration.
>
> On the topic of graphics engines, such as Unity and Unreal, the paper `A Review of Synthetic Image Data and Its Use in Computer Vision' indicates that the choice of engine is less impactful than the quality of 3D models. Hence, our discussion emphasizes the significant role of 3D model creation in generating high-quality synthetic data, rather than the effects of different engines.
>
> In conclusion, the impact of various data generation strategies, as the reviewer mentioned, opens a wide range of possibilities. This board field cannot be thoroughly explored within one paper. Our current work provides in-depth discussions of the widely used strategies and highlights potential future directions. It will promote further work and discussions around the effective use of synthetic data. We will continue to investigate these directions and add the aforementioned discussions to our revised paper.
>
> **Q2: It is recommended to measure the quality of the generated data and the impact of improving quality on the results.**
>
> Thank you for your valuable suggestion. Regarding the quality of synthetic data, the accuracy of labels is nearly 100\% due to the automated annotation process. This process involves generating images with annotation based on the IDs of the 3D person and vehicle models. Once these 3D identities are ready, a programmed system generates the annotations, thereby minimizing the risk of incorrect labelling. As a result, the label quality of our synthetic data is highly reliable, which is why we do not discuss the label quality much in the paper.
>
> To measure the quality of the generated images, we report the FID between real-world and synthetic data generated with Random and Attribute Descent (Attr. desc) methods in Table 6. The FID is widely used in evaluating the quality of GAN-generated data, which can help us understand the quality of synthetic data generated by different methods. For example, the FID between Market-1501 and synthetic data with Attr. desc is 83.24, lower than the FID of 104.78 for Market-1501 against synthetic data with random attributes. This difference in FID scores indicates that the data generated through Attr. desc is of higher quality than that generated using random attributes.
>
> Additionally, Section 6.2 of our paper discusses the effectiveness of content-level domain adaptation, which enhances the quality of synthetic data by mimicking the distribution of real data. Moreover, the results shown in Fig. 3 highlight the impact of data quality improvement on re-ID results. Synthetic data generated using attribute descent demonstrate improved re-ID results compared to randomly generated data.
>
> Following your recommendation, we will include more detailed discussions on the impact of data quality in the revised version of our paper.
>
> **Q3: The title of this paper is biased. It focuses on the person/vehicle re-identification, while the title uses object re-identification.**
>
> Thank you for the helpful reminder. We acknowledge the point of the reviewer that the current scope of the Alice project primarily centres on person/vehicle re-identification. We will revise the paper's title to more accurately reflect its current focus.

---

> > ### Author Response · Authors · 2023-11-22
> > **Friendly reminder**
> >
> > Dear Reviewer dCYN,
> >
> > This is a very gentle reminder that the discussion period is coming close to the end. Your comments and suggestions have been vital for helping improve the quality of our paper, which are greatly appreciated. We believe that we have addressed your concerns through our response. We do hope to have your feedback and look forward to answering any additional questions you have.
> >
> > Thank you very much.
> >
> > Best regards,
> >
> > The authors

---

### Meta-Review · Area_Chair_jXwT · 2023-12-03

**Metareview:**

This submission receives the following scores: 6, 8, 6, 6. All the reviewers agree to accept this submission.

The submission introduces the Alice Benchmarks, designed for advancing research in synthetic to real domain adaptation, specifically in the context of person and vehicle re-identification (re-ID). This benchmark aims to train models from synthetic data that are effective in real-world scenarios. The authors have set up an online server for evaluation and plan to expand the benchmarks to cover more objects in the visual world.

Considering the strengths ("Why Not Lower" below) and weaknesses ("Why Not higher" below), and the authors' response to reviewers' concerns, I recommend accepting this submission. It makes a fair contribution to the field, and while there are areas for improvement and expansion, the foundational work presented is sound and promising. The Alice Benchmarks address a current need in domain adaptation research and provide a platform for future advancements. The commitment to expanding the scope of the project is particularly encouraging, suggesting a long-term impact on the field.

**Justification For Why Not Higher Score:**

- Limited Scope in Current Form: The paper, while significant in its contribution, primarily focuses on person and vehicle re-ID. This narrower scope may limit its immediate impact across the broader domain of object re-identification.
- Data Generation Strategies: The paper could benefit from deeper analysis of different synthetic data generation strategies and their impact on domain adaptation results. Incorporating varied graphics engines, attribute distributions, and noise levels might provide richer insights.
- Data Quality and Label Accuracy: More thorough examination of the quality of generated data and labels, and their impact on the adaptation results, is recommended. This could enhance the reliability and applicability of the benchmarks.

**Justification For Why Not Lower Score:**

- Significant Contribution to Domain Adaptation Research: The benchmarks provide a valuable resource for the research community, facilitating advancements in the domain adaptation field. The inclusion of an online evaluation system adds to its utility and fairness.
- Comprehensive Experiments and Findings: The paper presents extensive experiments and findings, contributing to a deeper understanding of synthetic to real domain adaptation.
- Openness to Future Expansion: The authors express a commitment to expanding the Alice project, which shows potential for broader impact beyond its current scope.

---

### Decision · Program_Chairs · 2024-01-16

Accept (poster)